# Regression Models for Symbolic Interval-Valued Variables

**DOI:** 10.3390/e23040429

**Published:** 2021-04-06

**Authors:** Jose Emmanuel Chacón, Oldemar Rodríguez

**Affiliations:** 1National Bank of Costa Rica, San José 11501-2060, Costa Rica; jechaconch@bncr.fi.cr; 2School of Mathematics, CIMPA, University of Costa Rica, San José 11501-2060, Costa Rica

**Keywords:** regression, decision trees, random forest, boosting, K-nearest neighbors, support vector machines, neural networks, interval-valued variables, symbolic data analysis

## Abstract

This paper presents new approaches to fit regression models for symbolic internal-valued variables, which are shown to improve and extend the center method suggested by Billard and Diday and the center and range method proposed by Lima-Neto, E.A.and De Carvalho, F.A.T. Like the previously mentioned methods, the proposed regression models consider the midpoints and half of the length of the intervals as additional variables. We considered various methods to fit the regression models, including tree-based models, K-nearest neighbors, support vector machines, and neural networks. The approaches proposed in this paper were applied to a real dataset and to synthetic datasets generated with linear and nonlinear relations. For an evaluation of the methods, the root-mean-squared error and the correlation coefficient were used. The methods presented herein are available in the the RSDA package written in the R language, which can be installed from CRAN.

## 1. Introduction

Statistical and data mining methods have been developed mainly in cases in which variables take a single value. In real life, there are many situations in which the use of these type of variables can lead to an important loss of information or result in time-consuming calculations. In the case of quantitative variables, a more complete description can be achieved by describing an ensemble of statistical units in terms of interval data, that is the value taken by a variable is a closed interval in the real numbers.

It is especially useful when it is convenient to summarize large datasets in such a way that the resulting summary is of a manageable size and still maintains as much information as possible from the original dataset. For example, suppose we want to substitute the information of all transactions made by the owner of a credit card with a unique “transaction” summarizing all the original transactions. This is achieved thanks to the fact that this new transaction will have in its fields not only numbers, but also intervals defined by, for example, the minimum and maximum purchase.

The statistical treatment of interval-valued data was considered in the context of symbolic data analysis (SDA) introduced by E. Diday in [1], the objective of which is to extend the classic statistical methods to the study of more complex data structures that include, among others, interval-valued variables. A complete presentation on symbolic data analysis can be found in [2,3,4].

Research on SDA has focused primarily on unsupervised learning, with few contributions in the field of regression, which have been made mainly based on linear models. In this work, we explore nonlinear regression algorithms in conjunction with the center method and the center and range method for interval-valued data to try to improve the results involving classical linear regression for the center and center and range methods proposed in [5,6], respectively, and the extended lasso and ridge regression for interval-value data proposed in [7]. In this way, we are able to study how more sophisticated algorithms can improve the traditional models based on linear methods.

Furthermore, we extend the tool kit of algorithms available for regression on interval-valued data, taking advantage of the properties and the large number of algorithms for real-valued data. We explore the classical linear regression models, tree-based regression models (regression trees, random forest, and boosting), K-nearest neighbors regression, support vector machines regression, and regression using neural networks.

Section 2 gives a summarized presentation of the different regression models for real-valued data. Section 3 presents a summary of the center method and the center and range methods in the context of each regression model considered. Section 4 presents an experimental evaluation with a real dataset and simulated datasets, which evidences the differences and improvements among the models. Finally, Section 5 gives the concluding remarks based on the results.

## 2. Regression Methods

In this section, we provide a short review of the main regression methods and their mathematical formulation.

### 2.1. Classical Linear Regression Methods

We present only a summary of the classical linear regression model, while a complete presentation can be found in [8,9]. Linear regression models for decades have been some of the most important predictive methods in statistics; in fact, this continues today as one of the most important tools in statistics and data mining. The idea is, given an input vector $x^{t}=\left(x_{1},x_{2},\ldots ,x_{p}\right)$, where *p* is the number of variables and $x^{t}$ represents the transpose of *x*, we want to predict the response variable *y* through the following linear model:(1)$$\hat{y}={\hat{\beta }}_{0}+\sum_{j=1}^{p}x_{j}{\hat{\beta }}_{j},$$
where $\beta_{0}$ is called the intercept. If a constant one is included with vector *x* and $\beta_{0}$ in the coefficients vector β, the linear model can be written in vectorial form as a product as follows:(2)$$\hat{y}=x^{t}\hat{\beta },$$

To fit the linear model in the training data, the most popular estimation method is least squares. In this approach, we pick the coefficients β to minimize the residual sum of squares (RSS):(3)$$\mathrm{RSS}\left(\beta \right)=\sum_{i=1}^{n}{\left(y_{i}-x_{i}^{t}\beta \right)}^{2}.$$
where *n* is the number of observations in the dataset. RSS(β) is a quadric function; therefore, its minimum always exists. Note that it can be written as:(4)$$\mathrm{RSS}\left(\beta \right)={\left(y-X\beta \right)}^{t}\left(y-X\beta \right),$$
where *X* is an n×p matrix wherein each row is a vector in the training dataset; *y* is an *n* size vector (the output vector in the training dataset). It is well known that, if $X^{t}X$ is a nonsingular matrix, the solution is given by:(5)$$\hat{\beta }={\left(X^{t}X\right)}^{-1}X^{t}y.$$

The approximate value by this model for the component $x_{i}$ can be estimated as ${\hat{y}}_{i}=x_{i}^{t}\hat{\beta }$, and the fitted values for a new case $x^{t}=\left(1,x_{1},\ldots ,x_{p}\right)$ are given by $\hat{y}=x^{t}\hat{\beta }$.

In practice, there are various methods to find the coefficients $\hat{\beta }$, but the existing methods, and in particular the least squares method, are labor intensive and time consuming with large datasets, and others are not accurate enough with these kinds of datasets. A new non-iterative algorithm for identifying multiple regression coefficients based on the SGTMneural-like structure for the case of large volumes of data processing was proposed in [10], where the high efficiency of the method for the accuracy and speed in comparison with the existing methods was established.

Extensions to this model include shrinkage linear regression models, such as the ridge and lasso models. These models impose different types of regularization on the parameters: L2 regularization, used in ridge, and L1, used in lasso. A complete presentation of these models can be found in [8,11].

### 2.2. Tree-Based Regression Models

We now present a summary of the three main tree-based regression models. A complete presentation of these models can be found in [8].

#### 2.2.1. Regression Trees

There are two main steps for regression using decision trees: First, we begin by dividing the predictor space in $R^{n}$ into *J* non-overlapping regions $R_{1},\ldots ,R_{J}$; second, for every testing example that falls in $R_{j}$, we predict the response variable as the mean of the response variable over the training examples in $R_{j}$.

The regions $R_{j}$ could be of any geometrical shape, but for the sake of simplicity and interpretation, we restrict ourselves only to rectangular regions (boxes). Therefore, we are searching for a partition of the predictor space into boxes $R_{j}$ such that they minimize the RSS, which can be written as:$$RSS=\sum_{j=1}^{J}\sum_{i\in R_{j}}{\left(y_{i}-{\hat{y}}_{R_{j}}\right)}^{2}.$$

It is computationally unfeasible to consider every possible partition of the predictor space; for this reason, the method uses top-down and greedy recursive binary splitting. It is binary because every split of a predictor variable results in a division of the predictor space into two sets; it is top-down because it starts building the tree from the top to the leafs; and it is greedy because, at each step, the best split of the predictor is made without looking ahead and without picking a split that will lead to a lower RSS in some future step.

To construct the tree, first, we consider all the predictor variables and all the possible binary splits for their values. If the variable is numerical, all the possible values *s* in the range values of the variable are reconsidered, and if the variable is categorical, we consider all possible partitions in two sets of the values of the variable. We then select the variable and the split of that variable that leads to the greatest reduction in RSS. This produces two regions corresponding to two branches of the tree. We then continue looking for the best predictor and the best partition on each of the resulting regions, and the process ends when a stopping criterion is reached. It is common to build a large, complex tree and then prune it in order to reduce over fitting.

Once the regions $R_{1},\ldots ,R_{J}$ have been created, we can use it to predict the response for new examples as the mean of the response of the training observations in the region to which the new example belongs.

#### 2.2.2. Random Forest

The idea is to build a given number of regression trees on bootstrap sets (that is, obtain distinct datasets by repeatedly sampling with replacement observations from the original dataset) and use, in each tree, a random subset of *m* of the original predictor variables. Usually, *m* is taken to be $m=\sqrt{p}$, where *p* is the total number of variables.

In this case, the prediction of a new example is the mean over the predictions of the individual trees. The idea of random forest is to decorrelate the trees, thereby making the average of the resulting trees less variable.

#### 2.2.3. Boosting

In this method, the idea is to sequentially construct trees to repeatedly modify versions of the training data and the loss function, thereby producing a sequence of regression models, $G_{j}\left(x\right)$, whose predictions are then combined and weighted according to the error they produce.

Initially, all the *N* training examples $\left(x_{i},y_{i}\right)$ have the same weights $w_{i}=\frac{1}{N}$. In each following step of the training process, the data are modified by updating these weights $w_{i}$. At step *m*, those observations with a higher error by the model $G_{m-1}\left(x\right)$ induced at the previous step have their weights increased, whereas the weights are decreased for those that have a lower error, and those weights in turn are taken into account by the loss function. As a result of this weight actualization, each successive model in the process is then forced to concentrate on those training observations that present higher errors by previous models in the sequence.

### 2.3. K-Nearest Neighbors

Given a number *K* and a testing example $x_{0}$, we identify the set $N_{0}$ of the nearest *K* training examples to $x_{0}$. The prediction ${\hat{y}}_{0}$ of the response variable for $x_{0}$ is the mean of the response variable of the examples in $N_{0}$, that is:$${\hat{y}}_{0}=\frac{1}{K}\sum_{i\in N_{0}}y_{i}.$$

We can use any distance between examples, but it is recommended that the distance that minimizes the testing RSS is used. To select the appropriate number of neighbors *K*, it is recommended that cross-validation be used to compare the RSS of the resulting models, using $K=1,\ldots ,\sqrt{n}$, where *n* is the total number of examples.

More details on the K-nearest neighbors regression model were presented in [8].

### 2.4. Support Vector Machines

The support vector machines model for regression is an extension of the linear regression model that uses a metric that is less sensitive to outliers using a fixed margin to ignore the errors that are within this margin by means of a ϵ-insensitive function in the loss function and defines a hyperplane that is meant to adjust the data and define the prediction.

Given a threshold ϵ, the idea is to define a margin such that examples with residues within the margin do not contribute to the regression fitting, while residue examples outside the margin contribute proportionally to their magnitude. Therefore, the outlier observations have a limited effect, and the examples in which the model fits well do not have an effect on the model.

The loss function of this model is given by:$$LF\left(\beta \right)=C\sum_{i=1}^{n}L_{\epsilon }\left(y_{i}-{\hat{y}}_{i}\right)+\sum_{j=1}^{p}\beta_{j}^{2},$$
where *C* is the cost penalty, which penalizes large residuals, ${\hat{y}}_{i}=\beta_{0}+\beta_{1}x_{i1}+\cdots +\beta_{p}x_{i}p$, and $L_{\epsilon }\left(\xi \right)=\left\{\begin{array}{cc} |\xi |-\epsilon & \mathrm{si}\;|\xi |>\epsilon \\ 0 & \mathrm{si}\;|\xi |\leq \epsilon \end{array}\right.$ is the ϵ-insensitive function. We search for parameters ${\hat{\beta }}_{j}$ that minimize LF(β).

A complete presentation of this model can be found in [8].

### 2.5. Neural Networks

For a neural network model, the function *f* that approximates the true relation of the data $y=f^{*}\left(x\right)$ takes the form of a composition $f\left(x\right)=\left(f^{\left(n\right)}\circ f^{\left(n-1\right)}\circ \cdots \circ f^{\left(1\right)}\right)\left(x\right)$. In this chain structure, $f^{\left(i\right)}$ is called the *i*-th layer of the network, *n* is the depth of the network, and $f^{\left(n\right)}$ is the output of the network (which in the regression setup is a real-valued function), and the other layers are called hidden layers and are typically vector-valued functions. The neural network model is associated with a directed acyclic graph describing how the functions are composed together, and the idea of using many layers of vector-valued representations is that each one can learn distinct specific patterns in the data.

The training examples specify directly what the output layer must do at each point x; that is, it must produce a value that is close to the true value *y*. The behavior of the hidden layers is not directly specified by the training data; instead, the learning algorithm must decide how to use these layers to best implement an approximation of the true value *y*.

Neural networks are usually trained using stochastic gradient descent, which involves computing the gradients of complicated functions, and the back-propagation algorithm is used to efficiently compute these gradients.

Full details of the mathematical formulation of the neural network model can be found in [8,12].

## 3. Regression Models for Symbolic Interval-Valued Variables

In this section, we summarize the center method and the center and range method; a complete presentation of which can be found in [5,6,13,14,15], respectively. We also propose an approach to the center method and the center and range method in the context of the other regression models considered.

### 3.1. Center Method

In the center method, the β parameters are estimated based on the interval’s midpoints. In this method, there are predictors $X_{1},\ldots ,X_{p}$ and a response to be predicted *Y*, all of which are interval valued. Therefore, *X* is an n×p matrix, where each row is a vector of components of the training dataset $x_{i}=\left(x_{i1},\ldots ,x_{ip}\right)$ with $x_{ij}=\left[a_{ij},b_{ij}\right]$, and each component of the *Y* variable is also an interval $y_{i}=\left[y_{Li},y_{Ui}\right]$.

We denote by $X^{c}$ the matrix with the interval’s midpoints of the matrix *X*, that is $x_{ij}^{c}=\left(a_{ij}+b_{ij}\right)/2$, and we denote by $y_{i}^{c}=\left(y_{Li}+y_{Ui}\right)/2$ the midpoints of *Y*. The idea of the center method is to fit a linear regression model over $X^{c}=\left({\left(x_{1}^{c}\right)}^{t},\ldots ,{\left(x_{n}^{c}\right)}^{t}\right){)}^{t}$ with ${\left(x_{i}^{c}\right)}^{t}=\left(1,x_{i1}^{c},\ldots ,x_{ip}^{c}\right)$ for i=1,…,n and $y^{c}={\left(y_{1}^{c},\ldots ,y_{n}^{c}\right)}^{t}$. If ${\left(X^{c}\right)}^{t}X^{c}$ is nonsingular from (Equation 5), we know that the unique solution for β is given by:(6)$$\hat{\beta }={\left({\left(X^{c}\right)}^{t}X^{c}\right)}^{-1}{\left(X^{c}\right)}^{t}y^{c}.$$

The value of the prediction for $y=[y_{L},y_{U}]$ for a new case $x=(x_{1},\ldots ,x_{p})$ with $x_{j}=\left[a_{j},b_{j}\right]$ is estimated as follows:(7)$${\hat{y}}_{L}={\left(x_{L}\right)}^{t}\hat{\beta }\;\;\;\;\mathrm{and}\;\;\;\;{\hat{y}}_{U}={\left(x_{U}\right)}^{t}\hat{\beta },$$
where ${\left(x_{L}\right)}^{t}=\left(1,a_{1},\ldots ,a_{p}\right)$ and ${\left(x_{U}\right)}^{t}=\left(1,b_{1},\ldots ,b_{p}\right)$.

### 3.2. Center and Range Method

With the center and range method, Lima Neto and De Carvalho fit the linear regression model for interval-valued variables by using the information contained in the midpoints and in the interval ranges, in order to improve the quality of the prediction of the center method. The idea is to fit two regression models, the first with the midpoint of the interval and the second with the ranges of those same intervals. Just like the center method, there are $X_{1},\ldots ,X_{p}$ predictors and a response *Y*, and all these variables are interval valued. Therefore, *X* is an n×p matrix, where each row is a vector of a component of the training dataset $x_{i}=\left(x_{i1},\ldots ,x_{ip}\right)$ with $x_{ij}=\left[a_{ij},b_{ij}\right]$, and each component of the variable *Y* is also an interval $y_{i}=\left[y_{Li},y_{Ui}\right]$.

To fit the first regression model, we proceed in the same way as in the center method, that is if we denote by $X^{c}$ the midpoint’s matrix ($x_{ij}^{c}=\left(a_{ij}+b_{ij}\right)/2$) and we denote by $y_{i}^{c}=\left(y_{Li}+y_{Ui}\right)/2$ the midpoints of *Y*, the center and range method fits a first linear regression model over $X^{c}=\left({\left(x_{1}^{c}\right)}^{t},\ldots ,{\left(x_{n}^{c}\right)}^{t}\right){)}^{t}$ with ${\left(x_{i}^{c}\right)}^{t}=\left(1,x_{i1}^{c},\ldots ,{\left(x_{n}^{c}\right)}^{t}\right){)}^{t}$ for i=1,…,n and $y^{c}={\left(y_{1}^{c},\ldots ,y_{n}^{c}\right)}^{t}$. In this case, if ${\left(X^{c}\right)}^{t}X^{c}$ is nonsingular, then we know that the unique solution for $\beta^{c}$ is given by:(8)$${\hat{\beta }}^{c}={\left({\left(X^{c}\right)}^{t}X^{c}\right)}^{-1}{\left(X^{c}\right)}^{t}y^{c}.$$

To fit the second regression model, half of the value of the range of each interval is used. For this, we denote by $X^{r}$ the matrix that contains in each component half of the interval ranges of the matrix *X*, i.e., $x_{ij}^{r}=\left(b_{ij}-a_{ij}\right)/2$, and we denote by $y_{i}^{r}=\left(y_{Ui}-y_{Li}\right)/2$ half of the interval-valued variable *Y*. The center and range method then fits a second linear regression model over $X^{r}=\left({\left(x_{1}^{r}\right)}^{t},\ldots ,{\left(x_{n}^{r}\right)}^{t}\right){)}^{t}$ with ${\left(x_{i}^{r}\right)}^{t}=\left(1,x_{i1}^{r},\ldots ,x_{ip}^{r}\right)$ for i=1,…,n and $y^{r}={\left(y_{1}^{r},\ldots ,y_{n}^{r}\right)}^{t}$. In this case, if ${\left(X^{r}\right)}^{t}X^{r}$ is nonsingular from Equation (Equation 5), we know that the solution for $\beta^{r}$ is given by:(9)$${\hat{\beta }}^{r}={\left({\left(X^{r}\right)}^{t}X^{r}\right)}^{-1}{\left(X^{r}\right)}^{t}y^{r},$$
so each case in the training dataset is represented by two vectors $w_{i}=\left(x_{i}^{c},y_{i}^{c}\right)$ and $r_{i}=\left(x_{i}^{r},y_{i}^{r}\right)$ for i=1,…,n. The prediction value for $y=[y_{L},y_{U}]$ for a new case $x=(x_{1},\ldots ,x_{p})$ with $x_{j}=\left[a_{j},b_{j}\right]$ is then estimated as follows:(10)$${\hat{y}}_{L}={\hat{y}}^{c}-{\hat{y}}^{r}\;\;\;\;\mathrm{and}\;\;\;\;{\hat{y}}_{U}={\hat{y}}^{c}+{\hat{y}}^{r},$$
with:(11)$${\hat{y}}^{c}={\left(x^{c}\right)}^{t}{\hat{\beta }}^{c}\;\;\;\;\mathrm{and}\;\;\;\;{\hat{y}}^{r}={\left(x^{r}\right)}^{t}{\hat{\beta }}^{r},$$
where ${\left(x^{c}\right)}^{t}=\left(1,x_{1}^{c},\ldots ,x_{p}^{c}\right)$ and ${\left(x^{r}\right)}^{t}=\left(1,x_{1}^{r},\ldots ,x_{p}^{r}\right)$.

This model cannot mathematically guarantee that ${\hat{y}}_{Li}\leq {\hat{y}}_{Ui}$ for all i=1,…n, a problem addressed by Lima Neto and De Carvalho in [6].

Extensions to this model include shrinkage linear regression models for symbolic interval-valued data, which involve a generalization of the ridge and lasso models for interval-valued data. A complete presentation of these models can be found in [7].

### 3.3. Tree-Based Regression for Symbolic Interval-Valued Variables

#### 3.3.1. Regression Trees Center Method

We begin by dividing the predictor space of interval midpoints in $R^{n}$ into *J* non-overlapping regions $R_{1}^{c},\ldots ,R_{J}^{c}$.

We search for a partition of the predictor space into boxes $R_{j}^{c}$ such that they minimize $RSS^{c}$, which can be written as:$$RSS^{c}=\sum_{j=1}^{J}\sum_{i\in R_{j}^{c}}{\left(y_{i}^{c}-{\hat{y}}_{R_{j}}^{c}\right)}^{2}.$$

To construct the tree, first, we consider all the predictor variables and all the possible binary splits for their values. If the variable is numerical, all the possible values *s* in the range values of the variable are reconsidered, and if the variable is categorical, we consider all possible partitions in two sets of the values of the variable. We then select the variable and the split of that variable, which leads to the greatest reduction in $RSS^{c}$. This produces two regions corresponding to two branches of the tree. We then continue looking for the best predictor and the best partition on each of the resulting regions, and the process ends when a stopping criterion is reached.

Once the regions $R_{1}^{c},\ldots ,R_{J}^{c}$ have been created, the model can be used to predict the response of a new example $x=(x_{1},...,x_{p})$ with $x_{j}=\left[a_{j},b_{j}\right]$ as:$${\hat{y}}_{L}=\sum_{j=1}^{J}c_{j}\;1_{\left\{x_{L}\in R_{j}^{c}\right\}}\;\;\;\;\mathrm{and}\;\;\;\;{\hat{y}}_{U}=\sum_{j=1}^{J}c_{j}\;1_{\left\{x_{U}\in R_{j}^{c}\right\}},$$
where $c_{j}$ is the mean of the response centers of the training observations in $R_{j}^{c}$, $x_{L}=\left(a_{1},...,a_{p}\right)$ and $x_{U}=\left(b_{1},...,b_{p}\right)$.

#### 3.3.2. Regression Tree Center and Range Method

We begin by dividing the predictor space of interval midpoints in $R^{n}$ into *J* non-overlapping regions $R_{1}^{c},\ldots ,R_{J}^{c}$, and by dividing the predictor space of interval ranges in $R^{n}$ in *L* non-overlapping regions $R_{1}^{r},\ldots ,R_{L}^{r}$.

In this case, we are searching for a partition of the predictor space of centers and ranges into boxes $R_{j}^{c}$ and $R_{j}^{r}$ such that they minimize the $RSS^{c}$ and $RSS^{r}$, respectively, which can be written as:$$RSS^{c}=\sum_{j=1}^{J}\sum_{i\in R_{j}^{c}}{\left(y_{i}^{c}-{\hat{y}}_{R_{j}}^{c}\right)}^{2}\;\mathrm{and}\;RSS^{r}=\sum_{j=1}^{L}\sum_{i\in R_{j}^{r}}{\left(y_{i}^{r}-{\hat{y}}_{R_{j}}^{r}\right)}^{2}.$$

To construct the tree, first, we consider all the predictor variables and all the possible binary splits for their values. If the variable is numerical, all the possible values *s* in the range values of the variable are reconsidered, and if the variable is categorical, we consider all possible partitions in two sets of the values of the variable. We then select the variable and the split of that variable, which leads to the greatest reduction in $RSS^{c}$ and $RSS^{r}$. This produces two regions corresponding to two branches of the tree. We then continue looking for the best predictor and the best partition on each of the resulting regions, and the process ends when a stopping criterion is reached.

Once the regions $R_{1}^{c},\ldots ,R_{J}^{c}$, and $R_{1}^{r},\ldots ,R_{L}^{r}$, have been created, the model can be used to predict the response of a new example $x=(x_{1},...,x_{p})$ with $x_{j}=\left[a_{j},b_{j}\right]$ as:$${\hat{y}}_{L}={\hat{y}}^{c}-{\hat{y}}^{r}\;\;\;\;\mathrm{and}\;\;\;\;{\hat{y}}_{U}={\hat{y}}^{c}+{\hat{y}}^{r},$$
with:$${\hat{y}}^{c}=\sum_{j=1}^{J}c_{j}^{c}\;1_{\left\{x^{c}\in R_{j}^{r}\right\}}\;\;\;\;\mathrm{and}\;\;\;\;{\hat{y}}^{r}=\sum_{j=1}^{J}c_{j}^{r}\;1_{\left\{x^{r}\in R_{j}^{r}\right\}},$$
where $c_{j}^{c}$ and $c_{j}^{r}$ are the the means of the response centers and ranges of the training observations in $R_{j}^{c}$ and $R_{j}^{r}$, respectively.

#### 3.3.3. Random Forest Center Method

In this method, the idea is to build a given number *M* of regression trees, $T_{j}^{c}$, on bootstrap sets of the center data, using in each tree a random subset of *m* of the original predictor variables in $X^{c}$.

In this case, the prediction of a new example $x=(x_{1},...,x_{p})$ with $x_{j}=\left[a_{j},b_{j}\right]$ is the mean over the predictions of the individual trees:$${\hat{y}}_{L}=\frac{1}{M}\sum_{j=1}^{M}T_{j}^{c}\left(x_{L}\right)\;\;\;\;\mathrm{and}\;\;\;\;{\hat{y}}_{U}=\frac{1}{M}\sum_{j=1}^{M}T_{j}^{c}\left(x_{U}\right),$$
where $x_{L}=\left(a_{1},...,a_{p}\right)$ and $x_{U}=\left(b_{1},...,b_{p}\right)$.

#### 3.3.4. Random Forest Center and Range Method

With this method, the idea is to build a given number *M* of regression trees, $T_{j}^{c}$, on bootstrap sets of the center data and a given number *L* of regression trees, $T_{j}^{r}$, on bootstrap sets of the range data for each tree, using in each tree a random subset of *m* of the original predictor variables in $X^{c}$ and $X^{r}$.

In this case, the prediction of a new example $x=(x_{1},...,x_{p})$ with $x_{j}=\left[a_{j},b_{j}\right]$ is given by:$${\hat{y}}_{L}={\hat{y}}^{c}-{\hat{y}}^{r}\;\;\;\;\mathrm{and}\;\;\;\;{\hat{y}}_{U}={\hat{y}}^{c}+{\hat{y}}^{r},$$
with
$${\hat{y}}^{c}=\frac{1}{M}\sum_{j=1}^{M}T_{j}^{c}\left(x^{c}\right)\;\;\;\;\mathrm{and}\;\;\;\;{\hat{y}}^{r}=\frac{1}{L}\sum_{j=1}^{L}T_{j}^{r}\left(x^{r}\right).$$

#### 3.3.5. Boosting Center Method

We construct trees sequentially and repeatedly modify versions of the training center data, thereby producing a sequence of regression models, $G_{j}^{c}$, whose predictions are then combined and weighted according to the error they produce.

Initially, all the *N* training examples $\left(x_{i}^{c},y_{i}^{c}\right)$ have the same weights $w_{i}^{c}=\frac{1}{N}$. On each following step of the training process, the data are modified by updating these weights $w_{i}^{c}$. At step *m*, those observations with a higher error by the model $G_{m-1}^{c}\left(x\right)$ induced at the previous step have their weights increased, whereas the weights are decreased for those that have a lower error.

In this case, the prediction of a new example $x=(x_{1},...,x_{p})$ with $x_{j}=\left[a_{j},b_{j}\right]$ is the mean over the predictions of the individual trees:$${\hat{y}}_{L}=\sum_{j=1}^{M}\alpha_{j}^{c}\;G_{j}^{c}\left(x_{L}\right)\;\;\;\;\mathrm{and}\;\;\;\;{\hat{y}}_{U}=\sum_{j=1}^{L}\alpha_{j}^{c}\;G_{j}^{c}\left(x_{U}\right),$$
where $\alpha_{j}^{c}$ measures the error of the *j*-th model, $x_{L}=\left(a_{1},...,a_{p}\right)$ and $x_{U}=\left(b_{1},...,b_{p}\right)$.

#### 3.3.6. Boosting Center and Range Method

We construct trees sequentially and repeatedly modify versions of the training center and range data, thereby producing two sequences of regression models, $G_{j}^{c}$ and $G_{j}^{r}$, whose predictions are then combined and weighted according to the error they produce.

Initially, all the *N* center training examples $\left(x_{i}^{c},y_{i}^{c}\right)$ have the same weights $w_{i}^{c}=\frac{1}{N}$, and the same applies to the range training examples. In each following step of the training process, the data are modified by updating these weights $w_{i}^{c}$ and $w_{i}^{r}$. At step *m*, those observations with a higher error by the model $G_{m-1}^{c}\left(x\right)$ and $G_{m-1}^{r}\left(x\right)$ induced at the previous step have their weights increased, whereas the weights are decreased for those that have a lower error.

In this case, the prediction of a new example $x=(x_{1},...,x_{p})$ with $x_{j}=\left[a_{j},b_{j}\right]$ is given by:$${\hat{y}}_{L}={\hat{y}}^{c}-{\hat{y}}^{r}\;\;\;\;\mathrm{and}\;\;\;\;{\hat{y}}_{U}={\hat{y}}^{c}+{\hat{y}}^{r},$$
with:$${\hat{y}}^{c}=\sum_{j=1}^{M}\alpha_{j}^{c}\;G_{j}^{c}\left(x^{c}\right)\;\;\;\;\mathrm{and}\;\;\;\;{\hat{y}}^{r}=\sum_{j=1}^{M}\alpha_{j}^{r}\;G_{j}^{r}\left(x^{r}\right),$$
where $\alpha_{m}^{c}$ measures the error of the *j*-th center model and $\alpha_{j}^{r}$ measures the error of the *j*th range model.

### 3.4. K-Nearest Neighbors Center Method

Given a number *K* and a testing example $x=(x_{1},...,x_{p})$ with $x_{j}=\left[a_{j},b_{j}\right]$ in the symbolic dataset, we identify the sets $N_{0}$ of the nearest *K* training examples to $x^{c}$. The prediction $\hat{y}$ of the response variable for *x* is given by:$${\hat{y}}_{L}=\frac{1}{K}\sum_{x_{i}\in N_{0}}{\left(y_{i}\right)}_{L}\;\;\;\;\mathrm{and}\;\;\;\;{\hat{y}}_{U}=\frac{1}{K}\sum_{x_{i}\in N_{0}}{\left(y_{i}\right)}_{U}.$$

### 3.5. K-Nearest Neighbors Center and Range Method

Given the numbers $K^{c}$ and $K^{r}$ and a testing example *x* in the symbolic dataset, we identify the sets $N^{c}$ and $N^{r}$ of the nearest $K^{c}$ and $K^{r}$ training examples to $x^{c}$ and $x^{r}$, respectively. The prediction $\hat{y}$ of the response variable for *x* is given by:$${\hat{y}}_{L}={\hat{y}}^{c}-{\hat{y}}^{r}\;\;\;\;\mathrm{and}\;\;\;\;{\hat{y}}_{U}={\hat{y}}^{c}+{\hat{y}}^{r},$$
with:$${\hat{y}}^{c}=\frac{1}{K^{c}}\sum_{x_{i}^{c}\in N^{c}}y_{i}^{c}\;\;\;\;\mathrm{and}\;\;\;\;{\hat{y}}^{r}=\frac{1}{K^{r}}\sum_{x_{i}^{r}\in N_{r}}y_{i}^{r}.$$

### 3.6. Support Vector Machines Center Method

Given a threshold $\epsilon^{c}$, the idea is to define a margin such that examples with residues within the margin do not contribute to the regression fitting, while residue examples outside the margin contribute proportionally to their magnitude.

The loss function of this model is given by:$$LF^{c}\left(\beta \right)=C\sum_{i=1}^{n}L_{\epsilon }\left(y_{i}^{c}-{\hat{y}}_{i}^{c}\right)+\sum_{j=1}^{p}\beta_{j}^{2},$$
where:$$L_{\epsilon }\left(\xi \right)=\left\{\begin{array}{cc} \left|\xi \right|-\epsilon^{c} & \mathrm{si}\;|\xi |>\epsilon^{c}\\ 0 & \mathrm{si}\;|\xi |\leq \epsilon^{c} \end{array}\right..$$

We search for parameters ${\hat{\beta }}^{c}$ that minimize LF(β), so the value of the prediction for $y=[y_{L},y_{U}]$ for a new case $x=(x_{1},\ldots ,x_{p})$ with $x_{j}=\left[a_{j},b_{j}\right]$ is estimated as follows:$${\hat{y}}_{L}={\left(x_{L}\right)}^{t}{\hat{\beta }}^{c}\;\;\;\;\mathrm{and}\;\;\;\;{\hat{y}}_{U}={\left(x_{U}\right)}^{t}{\hat{\beta }}^{c},$$
where ${\left(x_{L}\right)}^{t}=\left(1,a_{1},\ldots ,a_{p}\right)$ and ${\left(x_{U}\right)}^{t}=\left(1,b_{1},\ldots ,b_{p}\right)$.

### 3.7. Support Vector Machines Center and Range Method

Given thresholds $\epsilon^{c}$ and $\epsilon^{r}$, we define margins such that examples with residues within these margin do not contribute to regression fitting, while residue examples outside the margins contribute proportionally to their magnitude.

The loss function of the center and ranges models is given by:$$LF^{c}\left(\beta \right)=C\sum_{i=1}^{n}L_{\epsilon^{c}}\left(y_{i}^{c}-{\hat{y}}_{i}^{c}\right)+\sum_{j=1}^{p}\beta_{j}^{2}\;\mathrm{and}\;LF^{r}\left(\beta \right)=C\sum_{i=1}^{n}L_{\epsilon^{r}}\left(y_{i}^{r}-{\hat{y}}_{i}^{r}\right)+\sum_{j=1}^{p}\beta_{j}^{2},$$
where:$$L_{\epsilon }\left(\xi \right)=\left\{\begin{array}{cc} |\xi |-\epsilon & \mathrm{si}\;|\xi |>\epsilon \\ 0 & \mathrm{si}\;|\xi |\leq \epsilon \end{array}\right..$$

We search parameters ${\hat{\beta }}^{c}$ that minimize LF(β), so the value of the prediction for $y=[y_{L},y_{U}]$ for a new case $x=(x_{1},\ldots ,x_{p})$ with $x_{j}=\left[a_{j},b_{j}\right]$ is estimated as:$${\hat{y}}_{L}={\hat{y}}^{c}-{\hat{y}}^{r}\;\;\;\;\mathrm{and}\;\;\;\;{\hat{y}}_{U}={\hat{y}}^{c}+{\hat{y}}^{r},$$
with:$${\hat{y}}^{c}={\left(x^{c}\right)}^{t}{\hat{\beta }}^{c}\;\;\;\;\mathrm{and}\;\;\;\;{\hat{y}}^{r}={\left(x^{r}\right)}^{t}{\hat{\beta }}^{r},$$
where ${\left(x^{c}\right)}^{t}=\left(1,x_{1}^{c},\ldots ,x_{p}^{c}\right)$ and ${\left(x^{r}\right)}^{t}=\left(1,x_{1}^{r},\ldots ,x_{p}^{r}\right)$.

### 3.8. Neural Networks Center Method

We consider a neural network model on the center data, and the function $f^{c}$ that approximates the true relation of these data $y=f^{c}\left(x\right)$ takes the form of a composition $f^{c}\left(x\right)=\left(f_{n}^{c}\circ f_{n-1}^{c}\circ \cdots \circ f_{1}^{c}\right)\left(x\right)$.

In this case, the prediction of a new example $x=(x_{1},...,x_{p})$ with $x_{j}=\left[a_{j},b_{j}\right]$ is given by:$${\hat{y}}_{L}=\left(f_{n}^{c}\circ f_{n-1}^{c}\circ \cdots \circ f_{1}^{c}\right)\left(x_{L}\right)\;\mathrm{and}\;{\hat{y}}_{U}=\left(f_{n}^{c}\circ f_{n-1}^{c}\circ \cdots \circ f_{1}^{c}\right)\left(x_{U}\right),$$
where $x_{L}=\left(a_{1},...,a_{p}\right)$ and $x_{U}=\left(b_{1},...,b_{p}\right)$.

### 3.9. Neural Networks Center and Range Method

We consider two neural networks models, one for the center data and another for the range data. The function $f^{c}$ that approximates the true relation of these data $y=f^{c}\left(x\right)$ takes the form of a composition $f^{c}\left(x\right)=\left(f_{n}^{c}\circ f_{n-1}^{c}\circ \cdots \circ f_{1}^{c}\right)\left(x\right)$.

In this case, the prediction of a new example $x=(x_{1},...,x_{p})$ with $x_{j}=\left[a_{j},b_{j}\right]$ is given by:$${\hat{y}}_{L}={\hat{y}}^{c}-{\hat{y}}^{r}\;\;\;\;\mathrm{and}\;\;\;\;{\hat{y}}_{U}={\hat{y}}^{c}+{\hat{y}}^{r},$$
with:$${\hat{y}}^{c}=\left(f_{n}^{c}\circ f_{n-1}^{c}\circ \cdots \circ f_{1}^{c}\right)\left(x^{c}\right)\;\;\;\;\mathrm{and}\;\;\;\;{\hat{y}}^{r}=\left(f_{n}^{r}\circ f_{n-1}^{r}\circ \cdots \circ f_{1}^{r}\right)\left(x^{r}\right).$$

## 4. Experimental Evaluation

As done by Lima Neto and De Carvalho in [6], the evaluation of the results of these interval-valued regression models was carried out using the following metrics: the lower boundary root-mean-squared-error RMSE${}_{L}$, the upper boundary root-mean-squared-error RMSE${}_{U}$, the square of the lower boundary correlation coefficient $r_{L}^{2}$, and the square of the upper boundary correlation coefficient $r_{U}^{2}$.
(12)$${\mathrm{RMSE}}_{L}=\sqrt{\frac{\sum_{i=1}^{n}{\left(y_{Li}-{\hat{y}}_{Li}\right)}^{2}}{n}}\;\;\;\mathrm{and}\;\;\;{\mathrm{RMSE}}_{U}=\sqrt{\frac{\sum_{i=1}^{n}{\left(y_{Ui}-{\hat{y}}_{Ui}\right)}^{2}}{n}},$$
(13)$$r_{L}^{2}={\left(\frac{\mathrm{Cov}(y_{L},{\hat{y}}_{L})}{S_{y_{L}}S_{{\hat{y}}_{L}}}\right)}^{2}\;\;\;\mathrm{and}\;\;\;r_{U}^{2}={\left(\frac{\mathrm{Cov}(y_{U},{\hat{y}}_{U})}{S_{y_{U}}S_{{\hat{y}}_{U}}}\right)}^{2},$$
where $y_{i}=\left[y_{Li},y_{Ui}\right]$ and its corresponding prediction is ${\hat{y}}_{i}=\left[{\hat{y}}_{Li},{\hat{y}}_{Ui}\right]$, $y_{L}={\left(y_{L1},\ldots ,y_{Ln}\right)}^{t}$, ${\hat{y}}_{L}={\left({\hat{y}}_{L1},\ldots ,{\hat{y}}_{Ln}\right)}^{t}$, $y_{U}={\left(y_{U1},\ldots ,y_{Un}\right)}^{t}$, ${\hat{y}}_{U}={\left({\hat{y}}_{U1},\ldots ,{\hat{y}}_{Un}\right)}^{t}$; as is usual, Cov(Ψ,Φ) denotes the covariance among variables Ψ and Φ; $S_{\Psi }$ denotes the standard deviation of variable Ψ.

For the experimental evaluation, we used the following hyperparameters for the models.
Lasso and ridge: folds used in K-fold cross-validation to find the best tuning parameter λ: 10Regression trees: minimum number of observations in a node: 20; maximum depth of any node of the final tree: 10.Random forest: number of trees to grow: 500; number of variables randomly sampled at each split: $\frac{p}{3}$, where *p* is the total number of variables.Boosting: number of trees to grow: 500; shrinkage parameter applied to each tree in the expansion: 0.1; highest level of variable interactions allowed: 1.K-nearest neighbors: maximum number of neighbors considered: 20; kernel to use: triangular.Support vector machines: kernel to use: radial.Neural networks: number of layers: 1; number of neurons: 10; threshold as stopping criteria: 0.05; maximum steps for the training process: $10^{5}$.

All examples presented in this section were developed using the package RSDA (R for symbolic data analysis) constructed by the authors of this paper for applications in symbolic data analysis (see [16]). The datasets used in this section are also available in the RSDA package. A reproducible document with all the R code for the examples can be downloaded from http://www.oldemarrodriguez.com/publicaciones.html, accessed on 29 March 2021.

### 4.1. Cardiological Interval Dataset

To illustrate the application of the methods, we considered the data based on [17] and taken from [3], shown in Table 1, in which the pulse rate (Pulse), systolic blood pressure (Sys), diastolic blood pressure (Diast), Art1 and Art2 were recorded as an interval for each of the patients, where Art1 and Art2 are artificial variables added to the data table. The goal was to predict (Pulse). The dataset consisted of 44 rows and 5 attributes. Table 1 presents a glimpse of the dataset. To measure the performance of the models, we split the dataset into training and testing sets, using 70% and 30%, respectively.

After closer inspection of Table 2, it can be seen that the neural network had the best metrics for RMSE${}_{L}$, $r_{L}^{2}$, and $r_{U}^{2}$, and had the second-best RMSE${}_{U}$, the random forest center and range model being the best one. As a result, the neural network model was best for this dataset.

As mentioned before, the neural network center and range model consist of 2 neural networks with 1 layer and 10 neurons, and this selection was made to avoid overfitting the small dataset. This simple neutral network model was able to capture the relation in the data better than the other models, and there was a large difference in performance, even when compared with the second best.

In the previous table, the boosting method was omitted because the table is too short for the model to be fitted.

### 4.2. Monte Carlo Experiments and Applications

The usefulness of the methods proposed in this paper was evaluated through experiments with synthetic interval-valued datasets with different linear and nonlinear configurations.

#### 4.2.1. Synthetic Linear Symbolic Interval Datasets

We followed the approach of Lima Neto and De Carvalho in [14] and constructed a symbolic interval dataset with the center and range values of the intervals simulated according to a linear relationship.

Our goal was to construct a symbolic dataset with 375 rows and 4 variables. We used 250 rows as the learning set and 125 as the test set. We followed these steps to construct the synthetic dataset.
The random variable $X_{j}^{c}$ was uniformly distributed in [20,40].The random variable $Y^{c}$ was related to the random variables $X_{j}^{c}$ according to $Y^{c}=X^{c}\beta +\epsilon$, where $X^{c}=\left(1,X_{1}^{c},X_{2}^{c},X_{3}^{c}\right)$, $\beta =(\beta_{0},\beta_{1},\beta_{2},\beta_{3})$ with $\beta_{j}∼U\left[-10,10\right]$, and ϵ∼U[a,b].The random variable $Y^{r}$ was related to the variable $Y^{c}$ according to $Y^{r}=Y^{c}\beta^{*}+\epsilon^{*}$, in the same manner $X_{j}^{r}$ was related to $X_{j}^{c}$ according to $X_{j}^{r}=X_{j}^{c}\beta^{*}+\epsilon^{*}$, where $\beta^{*}∼U\left[0.5,1.5\right]$ and $\epsilon^{*}∼U\left[i,j\right]$.

The next table shows different configurations for the values of a,b and i,j.

As per Lima Neto and De Carvalho, these configurations in Table 3 took into account the error at the midpoints combined with the degree of dependence between the midpoints and ranges, with two levels of variability: high variability error U[−20,20], low variability error U[−5,5], a high degree of dependence U[1,5], and a low degree of dependence U[10,20].

We present the results for all the datasets.

For dataset D1, we were able to see that the lasso, ridge, and LM center and range models were the best.

For dataset D2, the lasso, ridge, and LM center and range models were the best, and the KNN showed the next best performance.

Once again, for dataset D3, the lasso, ridge, and LM center and range model were the best models, without any close competitors.

For dataset D4, the lasso, ridge, LM, and the boost center and range model were the best. Therefore, the boost model was able to perform well in this dataset.

Note that in Table 4, Table 5, Table 6 and Table 7, some models have NA on $r_{L}^{2}$ or $r_{U}^{2}$, and this is because for those models, either the prediction ${\hat{y}}_{L}$ or ${\hat{y}}_{U}$ had zero standard deviation, and thus, the square of the correlation coefficient cannot be calculated.

#### 4.2.2. Synthetic Nonlinear Symbolic Interval Datasets

We considered a synthetic interval-valued dataset with a nonlinear relation between the response variable and the explanatory variables for the midpoint and range of the intervals.

In order to test the predictive power of the proposed methods, we followed the approach of Lima Neto and De Carvalho in [18] and constructed a symbolic interval dataset with the values of the center and range of the intervals simulated according to a nonlinear relationship between the response variable and the explanatory variables.

Our goal was to construct a symbolic dataset with 3000 rows and 4 variables and compare the methods using a K-fold cross-validation scheme with 10 folds. We followed the following steps to construct the synthetic dataset.
The center random variables $X_{j}^{c}$ were uniformly distributed in the interval [−6,6]; that is, $X_{j}^{c}∼U\left[-6,6\right]$.The center random variable $Y^{c}$ was related to the random variables $X_{j}^{c}$ according to the logistic function:
$$Y_{i}^{c}=\frac{\theta_{0}^{c}}{\theta_{1}^{c}+e^{\theta_{2}^{c}X_{1i}^{c}+\cdots +\theta_{n+1}^{c}X_{ni}^{c}}}+\epsilon_{i}^{c},$$
where $X_{ji}^{c}$ and $Y_{i}^{c}$ are the *i*th entries of the variables $X_{j}^{c}$ and $Y^{c}$, respectively, $\theta_{0}^{c}∼U\left[1.9,2.1\right]$, $\theta_{1}^{c}∼U\left[2.9,3.1\right]$, and $\theta_{m}^{c}∼U\left[0.9,1.1\right]$ for m=2,...,n+1, and the error component $\epsilon_{i}^{c}$ is normally distributed, where $\epsilon_{i}^{c}∼N\left(0,0.05\right)$.The range random variables $X_{j}^{r}$ were uniformly distributed in the interval [1,4]; that is, $X_{j}^{c}∼U\left[1,4\right]$.The range random variable $Y^{r}$ was related to the random variables $X_{j}^{r}$ according to the exponential function:
$$Y_{i}^{r}=\theta_{0}^{r}+e^{-(\theta_{1}^{r}X_{1i}^{r}+\cdots +\theta_{n}^{r}X_{ni}^{r})}+\epsilon_{i}^{r},$$
where $X_{ji}^{r}$, $Y_{i}^{r}$ are the *i*th entries of the variables $X_{j}^{r}$, $Y^{r}$, respectively, $\theta_{0}^{c}∼U\left[0,0.5\right]$ and $\theta_{m}^{r}∼U\left[0.9,1.1\right]$ for m=1,...,n, and the error component $\epsilon_{i}^{r}$ is normally distributed, where $\epsilon_{i}^{r}∼N\left(0,0.01\right)$.

As per Lima Neto and De Carvalho, the previous configuration had a high nonlinearity degree at the midpoints and a high nonlinearity degree in the ranges, as defined by the error components.

We here present the results for the synthetic dataset.

We note from Table 8 that every CRM method outperformed all of the CM methods. On closer inspection, we noticed that the neural network CRM model had the best cross-validation performance among all methods, and we observed that the proposed nonlinear models generally had better evaluation metrics. The second best method was the KNN CRM model, followed by the SVM CRM model.

Table 9 shows the standard deviation of the methods among the 10 folds.

As we can see from Table 9, the standard deviation was low for all methods among the folds, which gave us confidence in our results.

## 5. Conclusions and Future Work

In this paper, we proposed 12 new methods of fit regression models to interval-valued variables, all based on the central idea of fitting regression models for the centers and ranges of the intervals and extending the ideas of the nonlinear methods for real-valued data. We presented new approaches to fit regression models for symbolic internal-valued variables, which were shown to improve and extend the center method the center and range method proposed by Lima Neto and De Carvalho in [6,13,14,18].

In the experimental evaluation, we found that the use of nonlinear methods greatly improved the prediction results in the regression problems. With the cardiological dataset, a simple neural network was able to radically improve the predictions in comparison to the other methods, especially when compared to those based on linear methods. In the Monte Carlo experiments, as expected, the linear models were the best when using synthetic linear symbolic interval datasets, and only the boosting center and range model was close in performance in one of the datasets. When using the synthetic nonlinear symbolic interval data, we observed the real power and advantages of the nonlinear framework for regression with interval-valued data; in particular, we saw the benefit of using neural networks, as this was once again the model that best captured the underlying structure of the data, when combined with the center and range approach. When comparing with the linear models, we saw that the neural networks center and range model had a $RMSE_{L}$ and a $RMSE_{U}$ of 0.051, which was more than half lower than the root-mean-squared-errors of those center and range models based on linear methods of 0.141; in a similar way, the neural network had a $r_{L}^{2}$ and a $r_{U}^{2}$ of 0.97, which was much higher than the square of the correlation coefficients of those based on linear methods of 0.77. In addition, we want to note that almost all of the proposed models outperformed the classical models based on linear approaches and that these results did not consider hyperparameter tuning in the methods, which in turn represents an opportunity to further improve the results.

Based on the results found, we not only achieved our goal of extending the tool kit of regression models for interval-valued datasets, that had focused on linear methods, but also, we demonstrated the predictive advantages of making the extension to nonlinear methods. This is relevant due to the fact that, in real-life applications, data rarely follow a linear structure.

The methods proposed, just as the original center method, have the problem explained by Lima Neto and De Carvalho in [6], which is that it cannot be mathematically guaranteed that ${\hat{y}}_{L}\leq {\hat{y}}_{U}$. In future work, we will apply the idea proposed in [6], which consists of generating the regression models using certain restrictions that will allow us to guarantee that the methods satisfy this restriction. This was not included in this paper because it was expected to cause confusion in the results, since it would not have been clear if improvements in predictions were due to the applications of shrinkage or to the application of restrictions in the regression methods.

## Figures and Tables

**Table 1 entropy-23-00429-t001:** Cardiological interval dataset. Syst, systolic; Diast, diastolic; Art, artificial.

	Pulse	Syst	Diast	Art1	Art2
1	[44, 68]	[90, 100]	[50, 70]	[6, 9]	[1, 6]
2	[60, 72]	[90, 130]	[70, 90]	[2, 9]	[4, 6]
3	[56, 90]	[140, 180]	[90, 100]	[9, 10]	[4, 7]
4	[70, 112]	[110, 142]	[80, 108]	[8, 9]	[5, 5]
5	[54, 72]	[90, 100]	[50, 70]	[5, 7]	[6, 9]
⋮	⋮	⋮	⋮	⋮	⋮
44	[65, 67]	[90, 150]	[78, 90]	[8, 9]	[7, 9]

**Table 2 entropy-23-00429-t002:** Performance of the methods in the cardiological interval dataset.

Method	RMSE${}_{L}$	RMSE${}_{U}$	$r_{L}^{2}$	$r_{U}^{2}$
LM CM	15.8464	13.7108	0.1744	0.2238
Ridge CM	16.8055	13.5504	0.2234	0.2456
Lasso CM	17.2616	13.9484	0.0358	0.1652
RT CM	18.6783	15.9769	0.0055	0.0765
RF CM	15.0495	13.9372	0.3548	0.2406
KNN CM	16.2914	12.9731	0.3370	0.4125
SVM CM	15.5756	13.4378	0.4358	0.4307
NNet CM	9.2535	13.9276	0.6602	0.5340
LM CRM	12.3913	13.5961	0.2989	0.2191
Ridge CRM	12.5081	13.0821	0.3689	0.2368
Lasso CRM	13.5452	13.8853	0.1654	0.1381
RT CRM	12.7732	13.6137	0.2623	0.1730
RF CRM	7.6566	7.3574	0.7988	0.8172
KNN CRM	9.8981	10.7309	0.5889	0.4518
SVM CRM	11.9789	12.5868	0.3324	0.2728
NNet CRM	5.6585	7.4140	0.9270	0.9010

CM: center method. CRM: center and range method. LM: linear regression. Ridge: ridge regression. Lasso: lasso regression. RT: regression trees. RF: random forest. KNN: K-nearest neighbors. SVM: support vector machines. NNet: neural networks.

**Table 3 entropy-23-00429-t003:** Configurations with the center and range with a linear relationship. D1, Dataset 1.

D1	ϵ∼U[−20,20]	$\epsilon^{*}∼U\left[1,5\right]$
D2	ϵ∼U[−20,20]	$\epsilon^{*}∼U\left[10,20\right]$
D3	ϵ∼U[−5,5]	$\epsilon^{*}∼U\left[1,5\right]$
D4	ϵ∼U[−5,5]	$\epsilon^{*}∼U\left[10,20\right]$

**Table 4 entropy-23-00429-t004:** Performance of the methods for the D1 dataset.

Method	RMSE${}_{L}$	RMSE${}_{U}$	$r_{L}^{2}$	$r_{U}^{2}$
LM CM	94.2771	97.2412	0.2129	0.9710
Ridge CM	66.2051	67.6800	0.2131	0.9711
Lasso CM	92.9969	95.7236	0.2129	0.9710
RT CM	222.4937	324.0148	NA	NA
RF CM	258.4727	313.7617	NA	NA
Boost CM	235.2453	286.0998	NA	NA
KNN CM	248.2890	295.5031	0.0731	0.5838
SVM CM	332.8712	376.9175	0.0434	0.0282
NNet CM	133.2041	134.9265	0.2086	0.7714
LM CRM	11.1395	22.2623	0.2756	0.9716
Ridge CRM	10.2776	24.5461	0.2791	0.9719
Lasso CRM	11.0720	22.2897	0.2760	0.9717
RT CRM	25.5772	52.6583	0.0401	0.8410
RF CRM	11.9748	34.3630	0.1905	0.9521
Boost CRM	13.9365	25.9654	0.1388	0.9613
KNN CRM	12.0253	28.2744	0.2130	0.9563
SVM CRM	11.6466	27.3321	0.1579	0.9571
NNet CRM	12.6617	25.0032	0.1415	0.9641

NA: number not available.

**Table 5 entropy-23-00429-t005:** Performance of the methods for the D2 dataset.

Method	RMSE${}_{L}$	RMSE${}_{U}$	$r_{L}^{2}$	$r_{U}^{2}$
LM CM	208.2321	211.0179	0.0391	0.8514
Ridge CM	182.7565	185.2648	0.0389	0.8501
Lasso CM	207.0609	209.7936	0.0391	0.8509
RT CM	92.9444	119.6915	NA	NA
RF CM	109.5064	127.3871	NA	NA
Boost CM	84.7520	108.0348	NA	NA
KNN CM	99.2145	120.9664	0.0331	0.5261
SVM CM	153.5649	161.4119	NA	0.0196
NNet CM	41.5299	76.7332	0.0221	0.2926
LM CRM	12.3272	22.9804	0.7451	0.8829
Ridge CRM	10.8678	23.4012	0.7399	0.8817
Lasso CRM	12.2189	23.0000	0.7453	0.8826
RT CRM	21.3481	35.0539	0.2794	0.7257
RF CRM	11.3682	27.7170	0.5811	0.8503
Boost CRM	14.3557	23.3277	0.6593	0.8799
KNN CRM	12.4306	24.2769	0.7061	0.8723
SVM CRM	13.3351	24.3496	0.6763	0.8685
NNet CRM	15.5105	24.7335	0.6063	0.8644

NA: number not available.

**Table 6 entropy-23-00429-t006:** Performance of the methods for the D3 dataset.

Method	RMSE${}_{L}$	RMSE${}_{U}$	$r_{L}^{2}$	$r_{U}^{2}$
LM CM	731.6593	731.9502	0.8309	0.9835
Ridge CM	713.0555	711.7151	0.8309	0.9835
Lasso CM	730.6541	730.8220	0.8311	0.9835
RT CM	429.9846	405.3798	0.0000	NA
RF CM	428.2619	391.7614	0.0194	NA
Boost CM	392.7227	354.0930	0.2566	NA
KNN CM	413.8011	364.5694	0.4734	0.4805
SVM CM	491.7221	468.3171	0.1116	0.1944
NNet CM	343.0967	302.1161	0.8369	0.6365
LM CRM	30.4423	29.5111	0.9709	0.1197
Ridge CRM	34.7499	27.7174	0.9707	0.1324
Lasso CRM	30.6647	29.3510	0.9709	0.1198
RT CRM	70.3581	49.0615	0.8417	0.0329
RF CRM	54.0816	28.2500	0.9458	0.0724
Boost CRM	37.4534	31.9037	0.9576	0.0635
KNN CRM	41.8289	32.3533	0.9563	0.0203
SVM CRM	37.0102	29.4140	0.9615	0.0598
NNet CRM	36.1038	34.2631	0.9597	0.0487

NA: number not available.

**Table 7 entropy-23-00429-t007:** Performance of the methods for the D4 dataset.

Method	RMSE${}_{L}$	RMSE${}_{U}$	$r_{L}^{2}$	$r_{U}^{2}$
LM CM	209.5566	210.1047	0.0041	0.9305
Ridge CM	165.3670	165.6865	0.0042	0.9304
Lasso CM	205.1182	205.6540	0.0041	0.9306
RT CM	273.0309	289.5941	NA	NA
RF CM	274.3768	301.9242	NA	NA
Boost CM	241.1328	259.7999	NA	NA
KNN CM	252.3822	275.0876	0.0040	0.2819
SVM CM	339.6949	354.9830	NA	0.0714
NNet CM	184.6237	212.1473	0.0022	0.3013
LM CRM	18.1846	18.8634	0.0012	0.9473
Ridge CRM	16.7686	20.6743	0.0012	0.9470
Lasso CRM	18.0800	18.9213	0.0012	0.9474
RT CRM	23.3177	29.6393	0.0116	0.8671
RF CRM	15.7857	25.8713	0.0098	0.9272
Boost CRM	17.6997	19.1021	0.0089	0.9466
KNN CRM	18.8748	22.6928	0.0002	0.9299
SVM CRM	19.0737	21.1123	0.0053	0.9362
NNet CRM	21.0460	21.8625	0.0019	0.9287

NA: number not available.

**Table 8 entropy-23-00429-t008:** Ten-fold cross-validation mean metrics.

Method	RMSE${}_{L}$	RMSE${}_{U}$	$r_{L}^{2}$	$r_{U}^{2}$
LM CM	0.6656	0.6656	0.7322	0.7306
Ridge CM	0.6493	0.6493	0.7322	0.7306
Lasso CM	0.6634	0.6634	0.7322	0.7306
RT CM	0.2766	0.2927	0.3274	0.2710
RF CM	0.2426	0.2667	0.3991	0.3599
Boost CM	0.1717	0.1757	0.6961	0.6962
KNN CM	0.2525	0.2687	0.3356	0.3091
SVM CM	0.2798	0.2907	0.1969	0.2017
NNet CM	0.2642	0.2715	0.2695	0.2764
LM CRM	0.1411	0.1408	0.7743	0.7746
Ridge CRM	0.1418	0.1415	0.7743	0.7746
Lasso CRM	0.1411	0.1408	0.7743	0.7746
RT CRM	0.1495	0.1495	0.7462	0.7453
RF CRM	0.0681	0.0683	0.9531	0.9525
Boost CRM	0.1428	0.1425	0.7684	0.7687
KNN CRM	0.0580	0.0581	0.9621	0.9619
SVM CRM	0.0617	0.0617	0.9570	0.9568
NNet CRM	0.0514	0.0519	0.9701	0.9694

**Table 9 entropy-23-00429-t009:** Ten-fold cross-validation standard deviation metrics.

Method	RMSE${}_{L}$	RMSE${}_{U}$	$r_{L}^{2}$	$r_{U}^{2}$
LM CM	0.0111	0.0092	0.0123	0.0197
Ridge CM	0.0109	0.0093	0.0123	0.0197
Lasso CM	0.0111	0.0092	0.0123	0.0197
RT CM	0.0093	0.0040	0.0376	0.0318
RF CM	0.0075	0.0050	0.0268	0.0206
Boost CM	0.0089	0.0071	0.0305	0.0211
KNN CM	0.0075	0.0047	0.0221	0.0225
SVM CM	0.0073	0.0069	0.0190	0.0359
NNet CM	0.0079	0.0049	0.0209	0.0212
LM CM	0.0055	0.0049	0.0144	0.0129
Ridge CM	0.0054	0.0047	0.0144	0.0129
Lasso CM	0.0055	0.0048	0.0144	0.0129
RT CM	0.0117	0.0115	0.0463	0.0464
RF CM	0.0036	0.0030	0.0051	0.0047
Boost CM	0.0056	0.0051	0.0144	0.0139
KNN CM	0.0019	0.0012	0.0026	0.0026
SVM CM	0.0014	0.0017	0.0024	0.0030
NNet CM	0.0023	0.0023	0.0024	0.0028

## Data Availability

Data sharing not applicable.

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
