# Peer review of "Regression Models for Symbolic Interval-Valued Variables"

_entropy, 2021, doi:10.3390/e23040429_

Round 1

Reviewer 1 Report

As an initial statement, I know the Symbolic Data Analysis literature. I have worked for many years on this specific topic, participating in the international conference on the discipline. This, to confirm that:

1. the topic is very interesting
2. it can be possible to apply the in big data analysis, which is a hot topic nowadays
3. there is a lack of methodologies and also software in this context

The work proposed is, in my opinion, fascinating and useful in this context and the context of big data analysis. At the same time, the work, in my opinion, has the potential to be very cited, considered the fact Symbolic Data Analysis is actually growing, and there is a lack of methods and software to be used.

For this reason, I think is very useful the availability of software. Of course, it is relevant to present the approaches using a simulation study or an application. Still, considering the work's potential relevance, I confirm that the work merits adequate attention by the Journal and the fair improvement, In my opinion, can be published. 

Author Response

Thank you very much for the comments. 

Reviewer 2 Report

Paper deals with important task. Authors proposed a new approaches to fit regression models for symbolic internal-valued variables.

Paper has great practical value.

It has a logical structure, all necessary sections. Paper is technically sound. Experimental section is good.

The proposed approach are logical, results are clear.

Suggestions:

  1. Introduction section should be extended using more clearly motivation of this paper.
  2. It would be good to add point-by-point the main contributions in the end of the Introduction section
  3. It would be good to add the reminder of this paper
  4. Related works section should be extended using non-iterative approaches for solving the stated task (SGTM neural-like structure and its modifications). For example 10.1007/978-3-030-20521-8_39
  5. Authors should add all optimal parameters for all investigated methods (for Table 2, 4, 5, 6, 7, 8....)
  6. Conclusion section should be extended using: 1) numerical results obtained in the paper; 2)  prospects for the future research.
  7. A lot of references are outdated. Please fix it using 3-5 years old papers in high-impact journals.

Author Response

Thank you very much for the comments and suggestions. 

  1. Introduction section should be extended using more clearly motivation of this paper. R/ This was taken into account and carried out.
  2. It would be good to add point-by-point the main contributions in the end of the Introduction section. R/ This was added in the motivation of the article, in the conclusions and briefly in this section so as not to repeat the same thing a lot.
  3. Related works section should be extended using non-iterative approaches for solving the stated task (SGTM neural-like structure and its modifications). For example 10.1007/978-3-030-20521-8_39. R/ This was taken into account and carried out.
  4. Authors should add all optimal parameters for all investigated methods (for Table 2, 4, 5, 6, 7, 8....). R/ The hyperparameters used for each model were indicated: RT, RF, KNN, SVM, NNet.
  5. Conclusion section should be extended using: 1) numerical results obtained in the paper; 2)  prospects for the future research. R/ Regarding 1) About numerical results certain comments were added regarding Table 8. Regarding 2) they were already included in the conclusion.
  6. A lot of references are outdated. Please fix it using 3-5 years old papers in high-impact journals. R/ We add and cite two papers from 2018 and 2019.

Reviewer 3 Report

“Residual sum of squares” acronym should be introduced in text before equation (3).

In section 2.3, authors should write section title instead of providing the acronym. The acronym should be referred in text.

In subsection 4.1, Art1, and Art2 should be explained.

Table 2 subtitle should be reviewed. Items are overlapped.

Most of the work seems to be based on Lima Neto and De Carvalho [11, 14]. What new contributions are brought by this work? These should be highlighted in the conclusion.

In subsection 4.2.2, on the 3rd paragraph, where is written “compared” should be written “compare”.

Author Response

Thank you very much for the comments and suggestions.

“Residual sum of squares” acronym should be introduced in text before equation (3). R/ The acronym was introduced in text before equation (3).

In section 2.3, authors should write section title instead of providing the acronym. The acronym should be referred in text. R/ It was done. 

In subsection 4.1, Art1, and Art2 should be explained. R/ They were explained. 

Table 2 subtitle should be reviewed. Items are overlapped. R/ It was fixed. 

Most of the work seems to be based on Lima Neto and De Carvalho [11, 14]. What new contributions are brought by this work? These should be highlighted in the conclusion. R/ It was wrote at the beginning in the conclusion. 

In subsection 4.2.2, on the 3rd paragraph, where is written “compared” should be written “compare”. R/ It was fixed. 

Round 2

Reviewer 2 Report

Dear Authors,

thank you for the improvement of your paper.

It can be accepted in current form